# Myotendinous Junction: Exercise Protocols Can Positively Influence Their Development in Rats

**DOI:** 10.3390/biomedicines10020480

**Published:** 2022-02-18

**Authors:** Jurandyr Pimentel Neto, Lara Caetano Rocha-Braga, Carolina dos Santos Jacob, André Neri Tomiate, Adriano Polican Ciena

**Affiliations:** Laboratory of Morphology and Physical Activity—LAMAF, Institute of Biosciences—IB, São Paulo State University—UNESP, 24A, No.1515, Rio Claro 13506-900, SP, Brazil; jurandyr.pimentel@unesp.br (J.P.N.); lara.rocha@unesp.br (L.C.R.-B.); carolina.jacob@unesp.br (C.d.S.J.); andre.tomiate@unesp.br (A.N.T.)

**Keywords:** myotendinous junction, basal lamina, morphometry, telocyte

## Abstract

The myotendinous junction (MTJ) is an interface that different stimuli alter their morphology. One of the main stimuli to promote alterations in the MTJ morphology is physical exercise. The present study aimed to investigate the morphology and molecular MTJ adaptations of biceps brachii muscle in adult *Wistar* rats submitted to different ladder-based protocols. Forty *Wistar* rats (90 days old) were divided into four groups: Sedentary (S), Climbing (C), Overload Climbing (OC), Climbing, and Overload Climbing (COC). The results of light microscopy demonstrated the cell and collagen tissue reorganization in the experimental groups. The sarcomeres lengths of different regions showed a particular development according to the specific protocols. The sarcoplasmic invaginations and evaginations demonstrated positive increases that promoted the myotendinous interface development. In the extracellular matrix, the structures presented an increase principally in the COC group. Finally, the immunofluorescence analysis showed the telocytes disposition adjacent to the MTJ region in all experimental groups, revealing their network organization. Thus, we concluded that the different protocols contributed to the morphological adaptations with beneficial effects in distinct ways of tissue and cellular development and can be used as a model for MTJ remodeling to future proteomic and genetic analysis.

## 1. Introduction

The myotendinous junction (MTJ) consists of a highly specialized interface where the connection between the sarcoplasmic membranes and the extracellular matrix collagen fibers occurs [1,2]. Its main function is the force transmission generated through the muscle intracellular contractile proteins to the connective tissue proteins, which results in the articular joint movement and determines the MTJ as a higher transmission force area of the locomotor apparatus [3,4].

The MTJ ultrastructure is characterized by long and thin projections, originating from the extracellular matrix, known as sarcoplasmic invaginations (finger-like processes) that penetrate and bind to the muscle belly, through the sarcoplasmic evaginations. The myofibrillar bundles at its distal end are involved by these “interdigitations” [1,5].

Morphologically, interdigitations increase the MTJ contact interface and allow the contractile force transmission from the skeletal striated muscle to the tendon [6]. The MTJ development is directly related to the interaction influence of myoblasts and extracellular matrix elements, as well as the practical interaction between muscle and tendon tissues [7].

This anatomical region remains under extensive investigation, mainly regarding its adaptations and functional repercussions. The risk of muscle tension is probably a confluence of many factors linked to the MTJ ultrastructural architecture and its tissue contact surface [5,8,9,10].

The MTJ ultrastructure presents a higher adaptive capacity with physical exercise, with an increase in its contact area and different development aspects of sarcoplasmic invaginations and evaginations interaction [4,11]. MTJ showed structural changes in this region correlated to moderate aerobic training, a consequence of a higher number of tendon bifurcations and interdigitations, in addition to its relation to the sarcomeres interface, denominated proximal (PS), and distal (DS) sarcomeres [11,12].

The MTJ demonstrated a considerable increase in the length and thickness of its invaginations and evaginations, the collagen support area, and a reduction in the basal lamina after the ladder-based exercise, with and without overload. Furthermore, the first telocyte niche was reported in the MTJ interface after training [13].

It is possible to evidence changes in MTJ sarcomeres, for example in elderly female *Wistar* rats in menopause after swimming training [12]. The ladder-based protocols too demonstrated distinct characteristics in soleus and plantar muscles [11] with sarcomeres lengths’ changes as well as in the tibialis anterior and soleus muscle in an obesity experimental model [14].

Thus, this research aimed to investigate the MTJ remodeling and variation of described biceps brachii muscle submitted to different exercise stimuli, and the need for their general structural investigation and adaptation news.

## 2. Materials and Methods

### 2.1. Animals

Forty male *Wistar* rats, 90 days old, were divided into four Groups (n = 10): Sedentary (S)—did not participate in any experimental protocol; Climbing (C)—performed climbs without overload; Overload Climbing (OC)—performed climbs with additional progressive overload; Climbing and Overload Climbing (COC)—performed combined climbing protocols. During the experimental period, the animals were allocated to cages (33 × 40 × 16 cm, n = 4), under the conditions of temperature monitoring (23 ± 2 °C) and 12 h light/dark period, with food and water “*ad libitum*”. The study procedures applied were approved by the São Paulo State University Committee on Ethics in Animal Use of the Biosciences Institute (CEUA— No.2207, 28 March 2019).

#### 2.1.1. Climbing Protocol

The C Group performed an 8-week training protocol/l (3×/week) with 24 sessions in a vertical ladder-based (110 × 18 cm, 2 cm grids, 80° inclination) (Figure 1A). The sessions consisted of 9 climbs without additional overload (Figure 1B), where the animals were allowed to rest for 60 s in each [11,13].

#### 2.1.2. Overload Climbing Protocol

The OC Group performed an 8-week overload training protocol (3×/week). The sessions consisted of four to nine climbs with an additional and progressive overload, and the rats were allowed to rest for 120 s in each.

The 1st, 2nd, 3rd, and 4th climbs were performed, respectively, with 50%, 75%, 90%, and 100% of their weekly additional body mass overload, and in the subsequent climbs a 30 g of the extra progressive overload was added until the 9th climb was completed or exhaustion [11,13]. The load was fixed to the tail proximal region (Figure 1C).

#### 2.1.3. Climbing and Overload Climbing Protocol

The COC Group protocol consisted of an 8-weeks training protocol (3×/week) in a vertical ladder with and without an additional overload in alternate days. In the 1st session of the protocol, the COC Group executed four to nine climbs with additional and progressive overload. In the 2nd session, the rats realized nine climbs without additional load and so on until the 24th session was completed.

### 2.2. Light Microscopy—Image Area and Fractal Dimension

Three animals from each experimental group were euthanized (Anesthetic Overdose—Ketamine 100 mg/Kg, intraperitoneally), MTJ samples (3 mm^3^) from the right biceps brachii muscle were dissected and cryofixed in liquid nitrogen—196 °C. The samples were sectioned transversely at 10 µm (Cryostat—HM 505 E, MICROM^®^, Ramsey, MN USA), posteriorly dehydrated in increasing concentrations of alcohols, and stained with Hematoxylin-eosin (HE) [15] and Picro-Sirius Red (PSR) [16].

The images were acquired by a light microscope (Carl Zeiss^®^ Microimaging, Axioskop 2, Göttingen, Germany) were used only when they filled all the pixels with the analyzed tissue (n = 15) and it was calibrated using the ImageJ^®^ software (National Institutes of Health—NIH, Bethesda, MD, USA). The cell nuclei, collagen, and muscle fiber results—area and fractal dimension (FD)—were also obtained, after binarization.

The FD is used to quantify the organization of cellular elements through pixels distribution in the image space, without considering the image texture, its standardized value expressed will always be from 0 to 2, in which values that are close to 2 represent tissue or cell disorganization [17], the values were associated with the each analyzed variable area.

### 2.3. Transmission Electron Microscopy (TEM)

Four animals from each experimental group were euthanized (Anesthetic Overdose—Ketamine 100 mg/Kg, intraperitoneally), MTJ samples (3 mm^3^) from the right biceps brachii muscle were dissected and immersed in modified Karnovsky solution (containing 2.5% glutaraldehyde and 2% paraformaldehyde in sodium phosphate buffer solution, pH 7.4) for 48 h at 4 °C [18]. Afterward, the samples were post-fixed with 1% osmium tetroxide solution for 2 h at 4 °C, washed with saline solution, and the samples were submitted to dehydration in an increasing series of alcohols.

After dehydration, the samples were embedded in Spurr resin (Low Viscosity Embedding Media Spurs Kit Electron Microscopy Sciences, Sigma^®^ -Aldrich, Saint Louis, MO, USA). The sample blocks were trimmed to make 10 µm semi-thin sections (Leica^®^ Ultracut UCT, Göttingen, Germany), and stained with the 1% toluidine blue method. After selecting the interest area, ultra-thin 60 nm cuts were made, collected on 200 mesh copper grids (Sigma^®^-Aldrich, Saint Louis, MO, USA) and contrasted with a 4% uranyl acetate solution, washed in distilled water, contrasted in an 0.4% lead citrate aqueous solution, and then washed with distilled water [19].

The grids were examined using the Transmission Electron Microscope (Philips^®^ CM 100, Eindhoven, Netherlands) at the Laboratory of Electronic Microscopy Applied to Agricultural Research (NAP/MEPA-São Paulo University ESALQ/USP, São Paulo, Brazil). In the sarcomere analysis, we used 100 sarcomeres lengths for belly sarcomeres (BS), PS, and DS sarcomeres regions each one to establish their morphometric difference [20].

Moreover, the lengths (n = 75) and thickness (n = 100) of sarcoplasmic projections (invaginations and evaginations) (n = 75), also were measured the total MTJ interface (n = 10), which was delimited and standardized at 5 cm per image, at 48.4% amplification in the software, calibrated from the scale bars (ImageJ^®^—National Institutes of Health, Bethesda, MD, USA) [11].

The basal lamina thickness was measured in different regions along the MTJ (n = 50), and the collagen support layer thickness (CSL) (n = 50), and the transverse collagen fibers perimeter that form the CSL (n = 50) were too identified and measured in the extracellular matrix region [14].

### 2.4. Immunofluorescence

The MTJ biceps brachii samples (n = 3) were dissected and cryofixed in liquid nitrogen—196 °C. The samples were sectioned longitudinally at 20 µm (Cryostat—HM 505 E, MICROM^®^) and stained with CD34 for identifying the telocytes (CD34+) in the MTJ region. The muscle slides were washed in phosphate-buffered saline (PBS) for 5 min and incubated overnight at 4 °C with primary antibody CD34 (1:1000, IgG polyclonal, Invitrogen^®^, Oregon, OR, USA) diluted in PBS with 1% bovine serum albumin (BSA). After two washes in PBS, the slides were incubated with goat anti-rabbit secondary antibody Alexa Fluor 594 (1:1000, IgG, Invitrogen^®^, Oregon, OR, USA) diluted in PBS with BSA 1% solution for 1 h at room temperature.

The nuclei were stained with 4′,6-diamidino-2-phenylindole (DAPI) (Molecular Probes^®^, Eugene, USA). The histological sections were analyzed with a fluorescence microscope (Olympus BX61- NAP/MEPA—São Paulo University ESALQ/USP, São Paulo, Brazil) and the images were acquired at 200× magnification to visualize the muscle fiber disposition and the muscle-tendon interface [13].

## 3. Results

The results of the present study were analyzed and presented in topics related to each technique used, in addition, the summary of the quantitative results for a better understanding of the comparisons between groups was added in Table 1.

### 3.1. Light Microscopy—Image Area and Fractal Dimension

The FD of cell nuclei in the C Group was higher than in the OC and the COC group (*p* < 0.05), also the OC group presented a value higher than all the experimental groups (Figure 2A). The nuclei area in the C group had the higher nuclei area followed by OC, and COC had the lower area value (Figure 2B).

The FD of collagen tissue in the C group was higher than in the S group (*p* < 0.05) and lower compared to the OC group. The C group had a higher FD value compared to the experimental groups, besides the OC group was higher than the S and COC groups and lower than the C group. The COC group had a lower value and higher variation than all the experimental groups (Figure 2C).

The collagen tissue area in the C group was higher than the S and COC groups and lower than the OC group, in addition, the OC group had the higher collagen tissue area of the experimental groups principally compared to the S group (*p* < 0.05). The COC group had a larger area than the S group, but a lower area result compared to the other trained groups (Figure 2D).

The muscle fibers FD value in the C group was higher than the OC group (*p* < 0.005) and the COC group (*p* < 0.005). The OC group was lower than the S group (*p* < 0.05) and the C group (*p* < 0.05), but larger than the COC group, finally the COC group’s muscle fiber FD was lower compared to all the experimental groups (Figure 2E).

The C group had the lowest muscle fiber area, the OC group had the higher area value and the COC group was higher than the S and C groups but lower than the OC group (Figure 2F).

### 3.2. Transmission Electron Microscopy

#### 3.2.1. Sarcomeres

The BS lengths were longer in the C group compared to the S, OC, and COC groups (*p* < 0.0001). In addition, the OC group had longer sarcomere lengths than the S (*p* < 0.001) and COC groups and shorter than the C group, and finally, the COC group’s lengths were shorter than the S (*p* < 0.001), C, and OC groups (Figure 3C).

The PS lengths demonstrated in the C group were longer than the S, OC (*p* < 0.05), and COC groups. The OC group was shorter than S and C groups, and longer than the COC group. In addition, the COC group had the shortest lengths of the experimental groups (Figure 3C), demonstrating the effects in the MTJ region.

In relation to the DS lengths the C group had higher values than the S group (*p* < 0.01), and the OC group (*p* > 0.0001) and lower than the COC group. The OC group had the lowest values of the other experimental groups, finally, the COC group had higher values than the S (*p* > 0.0001), C, and OC (*p* > 0.001) groups (Figure 3C), also demonstrating the effects in the MTJ region.

#### 3.2.2. Myotendinous Junction

The lengths of the sarcoplasmic invagination were longer in the C group than in the S group, and shorter than in the OC and COC groups. The OC group presented higher lengths than the S (*p* < 0.0001), C (*p* < 0.0001), and COC (*p* < 0.05) groups. The COC group values were higher than the S (*p* < 0.0001) and C groups, and lower than OC group (Figure 4E).

The lengths of sarcoplasmic invaginations in the C group were longer than the S group (*p* < 0.05) and shorter than the OC and COC groups. The OC group had values higher than in the S (*p* < 0.0001), C (*p* < 0.001), and COC groups. The COC group had longer sarcoplasmic invaginations than the S and C groups and shorter than the OC group (*p* < 0.0001) (Figure 4E).

Regarding the invagination thickness, in the C group the values were higher than the S (*p* < 0.05) and OC groups and lower than the COC group. The OC group presented thickness lower than the S, C, and COC groups, and the COC group values were higher than those of the S (*p* < 0.01), C and OC (*p* < 0.05) groups (Figure 4F).

In the thickness results of sarcoplasmic evagination, the C group presented lower values than the S and COC groups and higher than the OC group. The OC group had thinner evaginations than the S (*p* < 0.0001), C (*p* < 0.05), and COC groups. The COC group was thinner than the S group, and larger than the C and OC groups (*p* < 0.05) (Figure 4F).

All the trained groups had an increase in the MTJ interface, however, did not present significant differences. The C group had a 19% longer interface, in the OC group it was 79.7% longer and in the COC group it was 28.9% longer than the S group. Compared to the C group, the OC group presented 51% higher values, and the COC group 8.3%. Among the groups that used additional overload during the protocols, the OC group values were 28.2% higher than the COC group (Figure 4G).

#### 3.2.3. Extracellular Matrix

The thickness of the basal lamina in the C group was larger than in the S and COC groups (*p* < 0.05) and shorter than in the OC group. The OC group had a higher thickness than the S, C, and COC groups (*p* < 0.01). The COC group had the lowest values than the other experimental groups (Figure 5D).

The thickness of the collagen support layer in the C group was higher than the S (*p* < 0.0001) and OC groups and lower than the COC group (*p* < 0.001), in the OC group the values were higher than the S group (*p* < 0.0001) and lower than the C and COC groups (*p* < 0.0001). In the COC group the collagen support layer was larger than the S (*p* < 0.0001), C (*p* < 0.001), and OC groups (*p* < 0.0001) (Figure 5E).

The collagen fiber perimeter in the C group was lower than the S and OC groups (*p* < 0.001) and longer than the COC group. In the OC group, the perimeter was lowest compared to the S, C, and COC *p* < 0.0001) groups. In the COC group, the perimeters were longer than the S (*p* < 0.0001), C, and OC (*p* < 0.0001) groups (Figure 5F).

## 4. Immunofluorescence

The immunostaining results with CD34+ and DAPI markers showed the telocytes’ disposition and their relationship with the MTJ region. In the S group, the telocytes are adjacent to the MTJ region principally aggregated in the tendon tissue. Already, in Group C, it was possible to identify telocytes in the MTJ region, especially close to its interface. In the OC group, it was possible to identify the presence of telocytes in the MTJ region, and between the muscle fibers close to this interface, they formed an integrated network that interconnected through their branches with the telopods. In the COC group, the telocytes were evidenced in a condensed network in the region of the myotendinous interface and in the tendon tissue (Figure 6).

## 5. Discussion

The present research demonstrated the cell nuclei organization, the collagen tissue area increase, the MTJ structures’ development, mainly in the muscular region after the overload protocol, and in the extracellular matrix, a highlight to the CSL increase and the telocytes network disposition in all the experimental groups.

In groups C and OC, there was an increase in the cell nuclei area in the muscle tissue, especially in the OC group (Figure 2B). Such changes were related to increased cellular activity and by a possible inflammatory process caused by exercise with overload, some aspects are demonstrated and associated with muscle recovery activity, the activation of satellite cells and muscle hypertrophy in humans [21,22].

The collagen tissue FD was higher in all groups that performed the protocol, especially in the OC group (Figure 2C). This indicates the accumulation of interstitial collagen associated with muscle development and, consequently, the increase in cell nuclei responsible for maintenance, and corroborates with a study that established FD to measure tissue health status and with fibrosis modulations caused by dystrophy [17,23].

The protocols contribute to sarcomeres becoming shorter with proximity to the MTJ interface (Figure 3C). The C and OC experimental models promoted shorter PS compared to the BS, and the DS lengths presented a higher length variation between the experimental groups. This can be associated with the development of the evagination and invagination, with thickness remodeling corroborating with different variations in the sarcomere ultrastructure, promoted by swimming [12,19] and ladder-based protocols [11].

The C group demonstrated a wide ramification of its invaginations (Figure 4B), corroborating studies that developed similar characteristics with exercise protocols too [4,24]. The experimental studies were subjected to exercises that prioritized aerobic capacity, and the findings complemented that, by showing the formation of sarcoplasmic invaginations’ ramifications, which involved evaginations and contributed to greater resistance of the muscle–tendon interface.

The invagination and sarcoplasmic evaginations lengths specifically in the C group corroborate with the data observed in a menopause model and swimming protocol [12] which presented a greater length of the sarcoplasmic invaginations than the evaginations. The mentioned study is significant in relation to the S group, where the MTJ demonstrated adaptations that contributed to their ultrastructure increase, even in exercise without overload [13].

In the OC group, there was high cellular activity at immunofluorescence analysis in the region close to the MTJ, and longer sarcoplasmic projections. These characteristics promoted a greater contact area, proven by the increase in the myotendinous interface already identified [25], consequently lowering susceptibility to injuries. Such modifications directly benefit the MTJ structure and can be considered in the future in human studies.

While in the OC group (Figure 3C), the increase in invagination lengths corroborated with the study which analyzed treadmill exercise in humans with progressive intensity [24], similar lengths’ results were presented, which indicate a greater adaptive response of the muscle tissue in the MTJ region associated with hypertrophy and the adaptation of the MTJ sarcomeres that compose the sarcoplasmic evagination.

Resistance exercise promoted morphological alterations that contributed to the muscle–tendon interface development [4,24,26,27]. This was presented in studies in different vertical ladder protocols [11,13] which demonstrated the proposed exercise’s beneficial effects with the MTJ ultrastructural development.

In relation to the basal lamina in Group C, there was attenuation in its thickness as in Group OC, results that differ from those found in studies [24] that used exercise on a treadmill, which showed an increase in this structure. This factor may be associated with the action of metalloproteinases that make up this region as different levels of laminin and type IV collagen [28], as they are responsible mainly due to the formation of the membrane skeleton at this region, in addition, that is associated with myogenic activity [29,30].

The basal lamina rearrangement in Figure 5A indicated cell activity also coming from its envelope. The satellite cells that presented, when activated control the homeostatic balance at the micro-injured tissues [31] and promoted changes in the membrane proteins as well as the different types of collagens in this region. In addition, they can influence their morphological adaptations, consequently, also the adjacent cells’ adaptations [32] mainly in muscle tissue [33].

The changes in SCL and its collagen fibers transversely arranged in Figure 5B,C with the protocols corroborates with the increase in collagen deposition [34]. The muscle adaptations, with different ladder-based protocols [13], and obesity [14], consequently promoted extracellular matrix remodeling and increased collagen production in different skeletal muscle regions [35]. In the MTJ such an association was evident mainly through the combination of protocols.

The perimeter increases in collagen fibers promoted a structural contribution to the extracellular matrix and, according to the contractile transmission force required in this region, promoted less risk of injuries and an increase in its region’s resistance [36,37].

The telocytes are particularly found in the interstitial area of some tissues such as smooth muscle [38], cardiac [39], in the tongue (skeletal striated muscle) [40], and in some organs such as the pancreas [41], testis [42], and liver [43] in different species. Telocytes have not yet been associated with exercise-related stimuli that may present its changes demonstrated in Figure 6 and in the future can be functionally analyzed in the human muscle-tendon interface.

The telocytes have already been identified in protocols of immobilization and aquatic remobilization [44], but their network and possible proliferation in the face of physical exercise was only demonstrated in exercise on a treadmill [45], albeit in a moderate way, compared to the vertical ladder exercises where we could observe their organization and diversification in relation to the protocols proposed.

Furthermore, the identification of a close telocyte niche and its extensions involving MTJ in all groups reinforces the fact that these cells are involved with the development of other cellular structures in this region. The finding of these cells at the MTJ may determine its niche of myoblast proliferation and the beginning of the post-inflammatory process arising from microlesions [46], mainly because in this region telocytes and satellite cells were found together in a similar vertical ladder experimental model [13].

## 6. Conclusions

We concluded that the adaptations are wide in the face of different vertical-based protocols, the present results demonstrated the morphological benefits to muscle and tendon tissue in different aspects, specifically at the myotendinous interface. The remodeling of MTJ is presented by an increase in its ultrastructure that supports and stabilizes the transmission of force through the region necessary for the different exercises.

Therefore, in the future, the experimental protocols in animals can be used as standards for this region’s development and for trends that encompass different genetic, proteomic, and molecular pathways acting during the process of MTJ remodeling. The telocytes network in all experimental groups indicated a link to muscle development caused by the exercise and can hereafter be associated with the myogenic and trophic factors acting on MTJ.

## Figures and Tables

**Figure 1 biomedicines-10-00480-f001:**
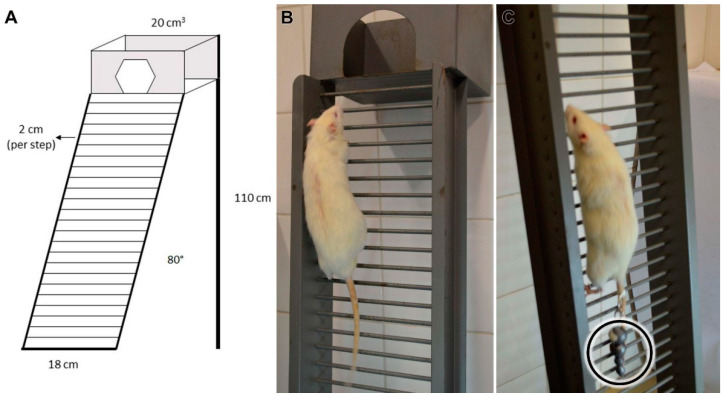
In the illustrative image we demonstrate the measurements of the main tool (**A**) the vertical ladder used in all experimental protocols; and the example of animals performing the (**B**) Climbing and (**C**) Overload Climbing (additional overload—circle) protocols that originate the association of the COC protocol.

**Figure 2 biomedicines-10-00480-f002:**
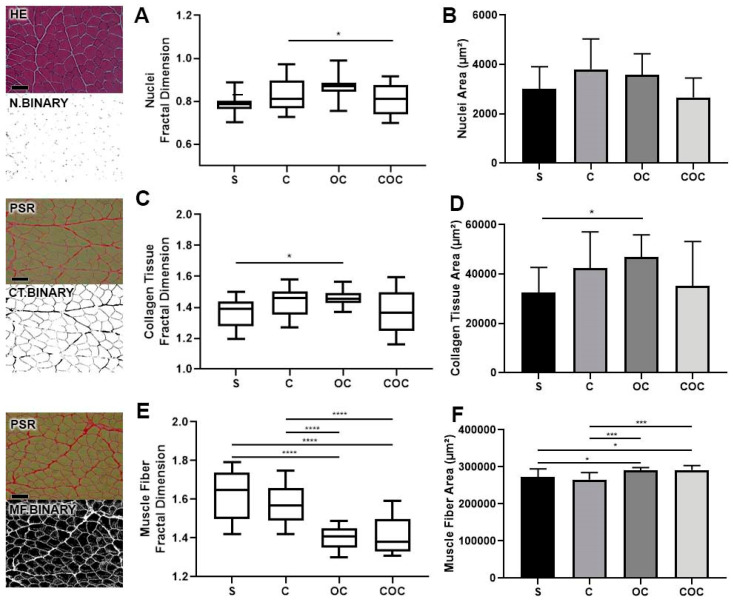
Fractal dimension data from the Hematoxylin and Eosin (HE) and Picro-Sirius (PSR) images to establish the stack and binary selection of nuclei (N.BINARY), collagen tissue (CT.BINARY), and muscle fiber (MF.BINARY) analysis from the Sedentary (S), Climbing (C), Overload Climbing (OC) and Climbing and Overload Climbing (COC) groups. Means and standard deviations: (**A**) Cell nuclei fractal dimension and (**B**) area; (**C**) Collagen tissue fractal dimension and (**D**) area; and the (**E**) muscle fiber fractal dimension and (**F**) area from all experimental groups (µm^2^). Data were statistically analyzed using the Kruskal–Wallis test of variance, with Dunn’s post-test (*p* < 0.05). Cell nuclei: * C ≠ COC (*p* < 0.05); Collagen tissue: * S ≠ OC (*p* < 0.05). Muscle Fiber: S ≠ OC e S ≠ COC * *p* > 0.05; C ≠ OC e C ≠ COC *** *p* > 0.005. **** S ≠ C, S ≠ COC, C ≠ OC, C ≠ COC (*p* < 0.0001). Bar: 200 µm Magnifications: 200×.

**Figure 3 biomedicines-10-00480-f003:**
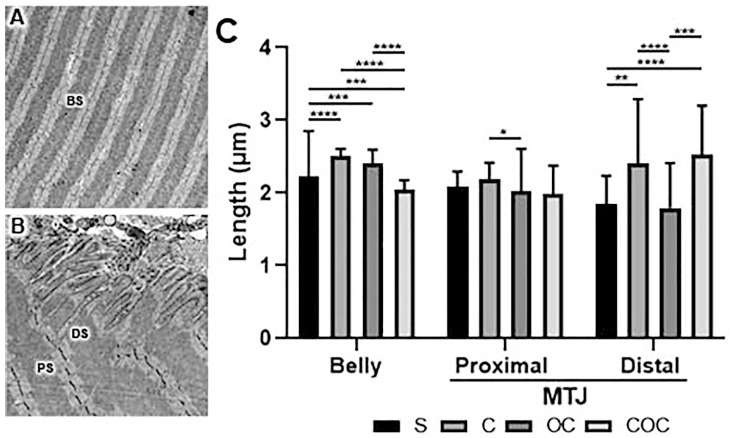
Ultrastructure image (TEM) of the (**A**) muscle belly sarcomere (BS); (**B**) proximal sarcomere (PS), and distal sarcomere (DS) of the MTJ; (**C**) Means and standard deviations of the muscle belly sarcomere lengths; (**B**) proximal sarcomere lengths, and distal sarcomere lengths of Sedentary (S), Climbing (C), Overload Climbing (OC), and Climbing and Overload Climbing (COC) groups. Data were statistically treated with Kruskal–Wallis analysis of variance, with Dunn’s post-test and significance level of *p* < 0.05. Belly: **** S ≠ C, C ≠ COC, OC ≠ COC (*p* < 0.0001); *** S ≠ OC, S ≠ COC (*p* < 0.001). MTJ: Proximal: * C ≠ OC (*p* < 0.05). Distal: ** S ≠ C (*p* < 0.01); **** S ≠ COC, C ≠ OC (*p* < 0.0001); *** OC ≠ COC (*p* < 0.001). Bar: 1 µm. Magnifications: (**A**) 10,000×; (**B**) 12,000×.

**Figure 4 biomedicines-10-00480-f004:**
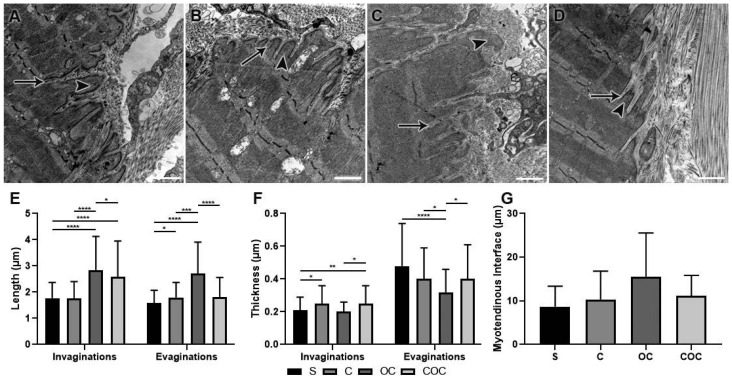
Ultrastructure images (TEM) of the myotendinous junction region with the sarcoplasmic invaginations (arrow) and evaginations (arrowhead) of the experimental groups (**A**) Sedentary (S), (**B**) Climbing (C), (**C**) Overload Climbing (OC), and (**D**) Climbing and Overload Climbing (COC). Means and standard deviations data were statistically treated using Kruskal–Wallis analysis of variance, with Dunn’s post-test (*p* < 0.05). (**E**) Invagination lengths: **** S ≠ OC, S ≠ COC, C ≠ OC (*p* < 0.0001); * OC ≠ COC (*p* < 0.05). Evagination lengths: * S ≠ C (*p* < 0.05); *** C ≠ OC (*p* < 0.001); **** S ≠ OC, OC ≠ COC (*p* < 0.0001); (**F**) Invagination thickness: * S ≠ C, OC ≠ COC (*p* < 0.05); ** S ≠ COC (*p* < 0.01). Evagination thickness: **** S ≠ OC (*p* < 0.0001); * C ≠ OC, OC ≠ COC (*p* < 0.05); (**G**) Myotendinous interface lengths of all experimental groups. Bars: 1 µm. Magnifications: 10,000×.

**Figure 5 biomedicines-10-00480-f005:**
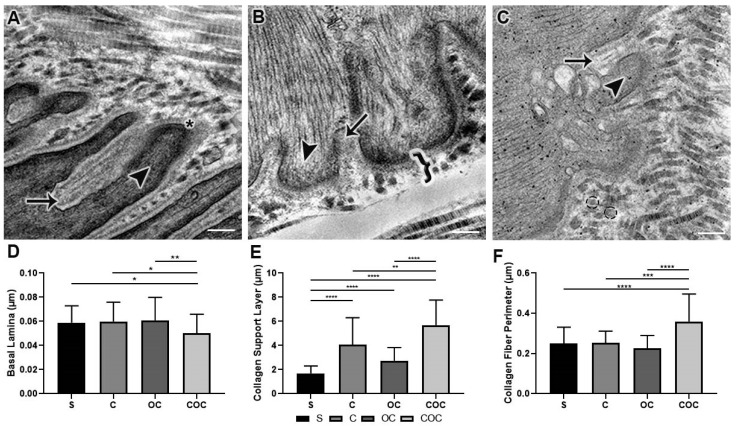
Ultrastructure images (TEM) of the basal lamina (star) that delimit the myotendinous interface and the extracellular matrix (**A**) where we can identify the sarcoplasmic invaginations (arrow) and evaginations (arrowhead); (**B**) in addition to the extracellular matrix, it was possible to identify and measure the (brace) Collagen Support Layer (CSL) and (**C**) (dashed circle) transverse collagen fiber that is present in the CSL of Sedentary (S), Climbing (C), Overload Climbing (OC) and Climbing and Overload Climbing (COC) groups. Means and standard deviations data were statistically treated using Kruskal–Wallis analysis of variance, with Dunn’s post-test (*p* < 0.05). (**D**) Basal Lamina: * S ≠ COC, OC ≠ COC (*p* < 0.05); ** OC ≠ COC (*p* < 0.01); (**E**) Collagen Support Layer: ** C ≠ OC (*p* < 0.01), **** S ≠ C, S ≠ OC, S ≠ COC, OC ≠ COC (*p* < 0.0001); (**F**) Collagen Fiber Perimeter: *** C ≠ COC (*p* < 0.001); **** S ≠ COC, OC ≠ COC (*p* < 0.0001). Bars: (**A**) 200 nm. (**B**) 100 nm (**C**) 50 nm. Magnifications: (**A**) 50,000×, (**B**) 80,000×, and (**C**) 30,000×.

**Figure 6 biomedicines-10-00480-f006:**
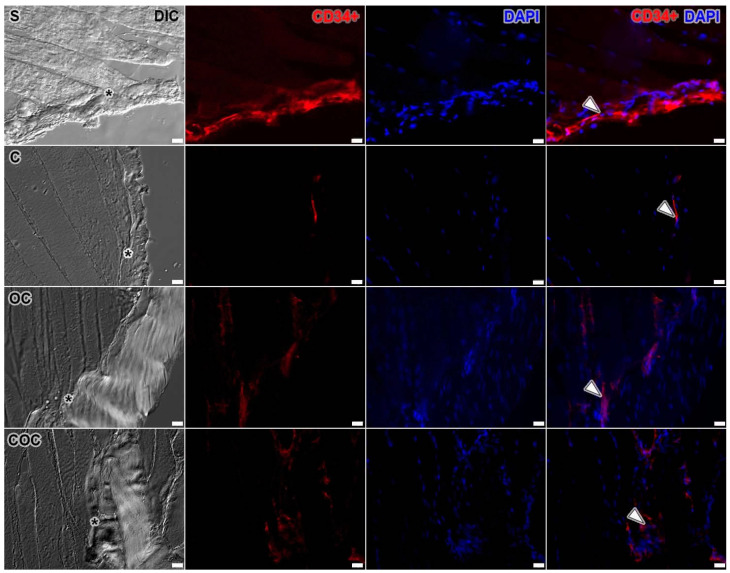
The MTJ interface (*) with Sedentary (S), Climbing (C), Overload Climbing (OC) and Climbing and Overload Climbing (COC) at DIC filter (gray), CD34+ immunofluorescence stain for telocytes identification (red), and DAPI (blue) for nuclei identification. The S group presents the identification of telocytes (arrow) in the MTJ region (*). In the C group, telocytes are in a tiny niche of the MTJ. In the OC group, the telocytes are present in specific clusters of each muscle fiber, whereas in the COC group the clusters are more distant. Bars: 20 µm. Amplification: 200×.

**Table 1 biomedicines-10-00480-t001:** Quantitative results were obtained in all experimental groups.

			Experimental Groups		
			C	OC	COC
Light Microscopy	Muscle	Nuclei FD (HE)	↑↑	↑↑↑	↑
		Nuclei Area (HE)	↑↑	↑	↓
		Collagen FD (PSR)	↑↑	↑	↑↑↑
		Collagen Area (PSR)	↑↑	↑↑↑	↑
		Muscle Fiber FD (PSR)	↓	↓↓	↓↓↓
		Muscle Fiber Area (PSR)	↓	↑	↑↑
Transmission Electron Microscopy	Sarcomeres	Belly	↑↑	↑	↓
		Proximal	↑	↓	↓↓
		Distal	↑	↓	↑↑
	Length	Sarcoplasmic Invaginations	↓	↑↑	↑
		Sarcoplasmic Evaginations	↑	↑↑↑	↑↑
	Thickness	Sarcoplasmic Invaginations	↑↑	↓	↑
		Sarcoplasmic Evaginations	↓↓	↓↓↓	↓
		Myotendinous Interface	↑	↑↑↑	↑↑
	Extracellular Matrix	Basal Lamina	↑	↑↑	↓
		Support Collagen Layer	↑↑	↑	↑↑↑
		Collagen Fiber Perimeter	↑	↓	↑↑

The analyses are a comparison to S group and classified by ↑ (reasonable increase); ↑↑ (considerable increase); ↑↑↑ (expressive increase); ↓ (reasonable reduction); ↓↓ (considerable reduction); ↓↓↓ (expressive reduction).

## Data Availability

All relevant data are within the paper.

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
