# Peer review of "Myotendinous Junction: Exercise Protocols Can Positively Influence Their Development in Rats"

_biomedicines, 2022, doi:10.3390/biomedicines10020480_

Round 1
Reviewer 1 Report
General Comments: The author’s attempt to investigate the impact of different ladder based exercise regimes on the MTJ remodeling and ultrastructure alterations of the biceps brachii muscle in male rats. A previous study cited by the authors investigated the same but in the plantaris muscle. While the experiments performed do seem to generate data required to investigate the author’s questions, the explanation drawn from the data are lacking in detail. The results section reads like a summary of the data presented and does not include what the data represents not what authors infer from the data. There are significant language issues where the message is unclear as well with some typographical, punctuation and grammatical errors. The authors also use some word choices that do not seem ideal for what this reviewer thinks the authors are trying to convey.
Specific Comments:
- In Figure 3, panels D, E, F and G are cut off. Please fix this.
- In Figure 5, immunofluorescence data from the right and bottom parts of the figure are cut off. Please include complete figure.
- The discussion section read more like the results section should and does not contain any thoughts from the authors regarding future directions, the impact of their work and how it may translate to human health. There are many sentences in this section that are tough to read and understand.
Author Response
Reviewer 1 Answer
We thank you for all the considerations and understand that several changes were necessary in all the topics of the article, in this way we revised and updated some terms for better understanding, mainly in the results and discussion. Also, we added a table in the results topic to inform the main adaptations of the protocols in an illustrative way.
- The figure panel was fixed and the corrected form was added.
- The figures were fixed and the corrected image was added.
- The results and discussion topics were reformulated and we hope that the changes have provided improvements in the understanding of our work.
Reviewer 2 Report
Well done reworking the manuscript.
I have some minor comments below here:
In Figure 1 and Figure 2: legends, please add information about S, C, OC and COC. These figures legends need to be representative of the manuscript as stand-alone information.
In Figure 3, section D and other parts are not visible; please correct these. Add necessary figure legends here too.
In Figure 4. Add necessary figure legends here too. The " * " in this figure could be written "star" as the other signs (like an arrow, arrowhead, brace...).
Figure 5, some parts are not visible; please correct these.
Page 9
Line 7-10. The introductory sentence is too long; please consider shortening it.
Line 14-18. This sentence can be separated into two.
Author Response
We thank you for the compliment and for the considerations, all were promptly resolved.
- We added the group information at the legends of the image
- The figure was fixed and we added the group information at the legend of the images.
- The figure was fixed and we added the group information at the legends of the images.
- The figure was fixed and we added the group information at the legends of the images.
- The sentence was reformulated.
- The sentence was reformulated.
Reviewer 3 Report
First, I would congratulate the research team that conducted this in vivo study for the original and remarkable idea.
In this work, the authors investigated how different exercise stimuli can affect myotendinous junctions in Wistar rats.
Despite the work overall is interesting, I have identified some points to be clarified and some corrections to make, particularly in results and discussion sections.
For these reasons, I propose a major revision, as follows:
- I advise researchers to add a photograph or a drawing of experimental phases to clarify the experiments carried out. The image should be inserted in materials and methods at 30/31 lines, page 2 (before protocols description).
- At the end of the protocol’s description, it would be appreciated to insert a table, which summarizes and makes a clear and concise comparison of the three types of exercises performed by rats.
- At page 3, it would be necessary to add numerations of the following paragraphs:
- 5 “Light Microscopy - Image Area and Fractal Dimension”
- 6 “Transmission Electron Microscopy (TEM)”
- 7 “Immunofluorescence”
Moreover, In the Result section, it is recommended to add a numeration of the following paragraphs:
- 1 “Light microscopy….”,
- 2 “Trasmission Electron….”
- 2.1 “Sarcomers”
- 2.2 “Myotendinous junction”
- 3 “Extracellular matrix”
- 4 “Immunofluorescence”
- At page 7, the figure 3 is not complete: the graphics are not complete, and the image “D” is not visible.
- At page 8, the caption of figure 4 is not formatted correctly.
- At page 8, the “immunofluorescence” paragraph is not formatted correctly.
- At page 9, the figure 5 is not complete and the caption is not formatted correctly.
- In the Discussion section, there are never references to figures reported in the Results section. I suggest discussing point by point referring to the graphs and images previously obtained.
- In the Discussion paragraph (at page 11, line 15), the authors explain the possible role of telocytes associated with exercise-related stimuli, but it’s not exhaustive and I recommend adding more information supported by literature data.
- In the conclusion part, at line 38, I would change the sentence: “Therefore, the experimental protocols in animals can be used as standards for…” in “therefore, in the future, the experimental protocols in animals could be used as standards for…”.
Author Response
We would like to thank you for the compliments and for the general understanding of our present work, mainly for the considerations to be improved in the text. We added an image that represents the main tool of our protocol and the base groups for carrying out the protocols, in addition we added a table with the quantitative data compared between the experimental groups for a better understanding of our variables and we made some point changes throughout the text to better association of objectives and data.
- We added a figure with some photos with our protocol models and a vertical ladder-base illustration with their measures.
- A table with quantitative data comparisons has been added to the end of the results section for better group comparison variables.
3) The numbers section was added and actualized.
4) The figures were corrected and now the images and graphics are complete.
5) We added some references of the figures and graphs in the discussion paragraphs for a better understanding of the associations between the results and associated work
6) There are still few studies associating exercise and telocytes in the actual literature. We have given a specific importance to our discovery, even so we added a few articles that correlate these variables and we hope that it has provided a better understanding of this relationship.
7) The conclusion topic was actualized.
Round 2
Reviewer 3 Report
The corrections requested have been made in a clear and exhaustive manner by the authors, allowing a better understanding of the article.